Quantitative study of the behavior of two broadcast spawners, the sea urchins Strongylocentrotus intermedius and Mesocentrotus nudus, during mass spawning events in situ

Zhadan Peter M. 1 pzhadan@poi.dvo.ru
http://orcid.org/0000-0002-9098-2102 Vaschenko Marina A. 2 mvaschenko@mail.ru
Permyakov Peter A. 1
1 Department of Geochemistry and Ecology of the Ocean, V. I. Il’ichev Pacific Oceanological Institute FEB RAS , Vladivostok, Primorsky Krai , Russia
2 Laboratory of Physiology, A.V. Zhirmunsky National Scientific Center of Marine Biology FEB RAS , Vladivostok, Primorsky Krai , Russia
Banaszak Anastazia
Electronic publication date: 2021 Apr 6
Publication date: 2021
Volume: 9
Electronic Location ID: e11058
Received 2020 Oct 13; Accepted 2021 Feb 12
Copyright: © 2021 Zhadan et al.
Copyright year: 2021
Copyright holder: Zhadan et al.
License: This is an open access article distributed under the terms of the Creative Commons Attribution License, which permits unrestricted use, distribution, reproduction and adaptation in any medium and for any purpose provided that it is properly attributed. For attribution, the original author(s), title, publication source (PeerJ) and either DOI or URL of the article must be cited.
License URL: https://creativecommons.org/licenses/by/4.0/

Keywords: Broadcast spawning, Echinoderms, Sea urchin reproduction, Sea urchin behavior, Reproductive adaptations, Chemosensation

Funding: State Programs of the Russian Federation 027120160006, АААА-А17-117030110038-5 and 115081110046 Russian Foundation for Basic Research 16-05-0008316a This work was supported by the State Programs of the Russian Federation (projects numbers 027120160006, АААА-А17-117030110038-5 and 115081110046) and the Russian Foundation for Basic Research (grant number 16-05-0008316a). The funders had no role in study design, data collection and analysis, decision to publish, or preparation of the manuscript.

==============================
Background

The spatial distribution of spawners and temporal parameters of spawning in motile invertebrates with external fertilization might influence reproductive success. However, to date, data on the prespawning and spawning behaviors of broadcast spawners in the field have been scarce and mostly qualitative. The present study was intended to clarify the behavioral adaptations of two sea urchin species, Strongylocentrotus intermedius and Mesocentrotus nudus, using quantitative analysis of their behavior during mass spawning events under natural conditions.

Methods

We analyzed in situ video recordings of sea urchin behavior obtained during six spawning seasons (2014–2019). The total number of specimens of each sea urchin species and the numbers of spawning males and females were counted. Quantitative parameters of sea urchin spawning (numbers of gamete batches, release duration of one gamete batch, time intervals between gamete batches and total duration of spawning) and movement (step length of spawners and nonspawners before and during spawning and changes in distances between males/nonspawners and females) were determined.

Results

For each species, 12 mass spawning events were recorded in which 10 or more individuals participated. The temporal dynamics of the numbers of males and females participating in mass spawning were well synchronized in both species; however, males began to spawn earlier and ended their spawning later than females. In both species, the most significant intersex difference was the longer spawning duration in males due to the longer pause between gamete batches. The total duration of gamete release did not differ significantly between sexes. The average duration of sperm release during mass spawning events was longer than solitary male spawning. Males and females showed significant increases in the locomotion rate 35 min before the start of spawning and continued to actively move during spawning. An increase in movement rate before spawning in males and females was induced by environmental factor(s). Nonspawners of both species showed increased locomotion activity but in the presence of spawning neighbors and less prominently than spawners. On a vertical surface, both echinoids moved strictly upward. On flat surfaces, males, females and nonspawners of both echinoids became closer during spawning.

Discussion

We showed that two sea urchin species with planktotrophic larvae display similar behavioral adaptations aimed at enhancing reproductive success. The high sensitivity of sea urchins, primarily males, to some environmental factors, most likely phytoplankton, may be considered a large-scale adaptation promoting the development of mass spawning events. The longer spawning duration in males and increased movement activity before and during spawning in both sexes may be considered small-scale adaptations promoting approach of males and females and enhancing the chances of egg fertilization.

Introduction

During the course of evolution, marine invertebrates with external fertilization (broadcast spawners) and feeding (planktotrophic) larvae have developed a number of reproductive adaptations ensuring successful reproduction and thereby maintaining the continuity of species. The broadcast spawning strategy involves the release of a large number of small eggs by females and sperm by males during synchronous (often annual) spawning events (see Olive, 1992; Wangensteen, Turon & Palacín, 2017 for review). It is generally believed that the synchronization of spawning of many individuals is provided by the ability of broadcast spawners to perceive certain environmental cues to determine the conditions appropriate for offspring development (see Mercier & Hamel, 2009; Thorson, 1950 for review). For echinoderms and sea urchins in particular, the external synchronizers are reported to be an increased level of phytoplankton (Bronstein, Kroh & Loya, 2016; Egea et al., 2011; Gaudette, Wahle & Himmelman, 2006; González-Irusta, De Cerio & Canteras, 2010; Himmelman, 1975; Starr, Himmelman & Therriault, 1993; Zhadan et al., 2016; Zhadan, Vaschenko & Ryazanov, 2018), an increase or decrease in water temperature (Byrne, 1990; Egea et al., 2011; Guillou & Lumingas, 1998; Himmelman et al., 2008; King, Hoegh-Guldberg & Byrne, 1994; Lamare & Stewart, 1998; Tsuji et al., 1989), photoperiod (Byrne et al., 1998), different phases of the moon cycle (Coppard & Campbell, 2005; Gaudette, Wahle & Himmelman, 2006; Iliffe & Pearse, 1982; Lessios, 1991; Mercier & Hamel, 2010; Zhadan et al., 2016), time of day (Zhadan, Vaschenko & Ryazanov, 2018) and the presence of released gametes in the environment (Reuter & Levitan, 2010; Unger & Lott, 1994). Although sea urchin spawning may be influenced by multiple natural factors, phytoplankton have been shown to be the most likely proximate cue for triggering spawning in some species (Gaudette, Wahle & Himmelman, 2006; Himmelman, 1975; Starr, Himmelman & Therriault, 1993; Zhadan, Vaschenko & Ryazanov, 2018). Moreover, under conditions of low phytoplankton abundance, spawning failure occurs in natural populations of the sea urchin Strongylocentrotus intermedius, followed by prolonged resorption of the unspawned eggs and sperm (Zhadan, Vaschenko & Almyashova, 2015; Zhadan et al., 2016). This phenomenon indicates that an external stimulus is extremely important for the reproduction of some species and suggests a high degree of impact of natural selection for the synchronization of spawning with environmental factors.

In motile species, behaviors aimed at reducing nearest-neighbor distances or increasing the likelihood of gamete survival and mixing may also favor fertilization success, especially at low population densities (Himmelman et al., 2008; Levitan, 2002, 2005; McCarthy & Young, 2004). A classic example of spawning organisms being located in close proximity is the pseudocopulation of ophiuroids and starfish (Himmelman et al., 2008; Slattery & Bosch, 1993; Tominaga, Nakamura & Komatsu, 2004). The temporal parameters of pseudocopulation differ among species. Males of dimorphic ophiuroids such as Ophiodaphne formata are constantly attached to females (Tominaga, Nakamura & Komatsu, 2004). Males and females of the sand sea star Archaster typicus form pairs 2 months before spawning, and males spawn only when females release eggs (Run et al., 1988). In other sea stars and ophiuroids, approaching of males and females occurs just before or during spawning. During echinoderm mass spawning events off the Mingan Islands (the northern Gulf of St. Lawrence, Canada), many instances of pseudocopulation of the ophiuroids Ophiopholis aculeata and Ophiura robusta and sea star Asterias vulgaris as well as of approaching individuals have been observed (Himmelman et al., 2008).

Climbing onto elevated surfaces before spawning is a typical behavior for many echinoderms (see Levitan, 1998; Mercier & Hamel, 2009 for review). In cases when both sexes move upward, such displacement evidently promotes approach of males and females. Spawning of starfish, ophiuroids and sea urchins in situ was observed on the tops of corals, stones, and even algae (Babcock et al., 1992; Gladstone, 1992; Hagman & Vize, 2003; Himmelman et al., 2008; Johnson & Ranelletti, 2017; McEuen, 1988; Minchin, 1992). The ophiuroids O. aculeata, O. robusta, Ophioderma rubicundum and Ophioderma squamosissimum leave their shelters in crevices before spawning and move to higher elevations (Hagman & Vize, 2003; Himmelman et al., 2008). Highly accelerated upward movement on a vertical surface immediately before and during spawning was also recorded in the sea urchin Strongylocentrotus intermedius (Zhadan, Vaschenko & Ryazanov, 2018). Some starfish and ophiuroids take a characteristic pose before spawning, raising the aboral disc above the bottom surface (Gladstone, 1992; Hagman & Vize, 2003; Hendler & Meyer, 1982; Himmelman et al., 2008). Many holothuroid species spawn with their anterior ends lifted off the substratum (Hendler & Meyer, 1982; McEuen, 1988). The behavior aimed at maximal elevation of the body in the water column seems to promote gamete dispersion and mixing and to increase gamete survival, mitigating their falling to the bottom and being wasted (Hendler & Meyer, 1982; Himmelman et al., 2008).

The above observations indicate that motile echinoderms are able to change their behavior before spawning events. Moreover, some species, including sea urchins, display an increase in locomotion rate before spawning (Himmelman et al., 2008; Zhadan, Vaschenko & Ryazanov, 2018), prompting the suggestion that this may be the first response of mature individuals to some exogenous and/or endogenous factors stimulating spawning. However, to date, only a few laboratory and field studies have attempted to quantitatively assess sea urchin movement in response to presumable spawning stimuli. During natural spawning events of the sea urchin Mesocentrotus (=Strongylocentrotus) franciscanus, which coincided with phytoplankton blooms, Levitan (2002) found no changes in the nearest-neighbor distances between males and females or between nonspawners. In laboratory experiments, phytoplankton, sperm and a combination of sperm and phytoplankton increased the rate of movement of sexually mature Lytechinus variegatus males and females up the wall of the experimental beaker, whereas only sperm and sperm+phytoplankton treatments induced vertical movement of sexually immature individuals (Reuter & Levitan, 2010). In the field experiments of McCarthy & Young (2004), sperm added to sea water did not influence the distance traveled per 1 h or the nearest-neighbor distance in the sea urchin L. variegatus with mature and immature gonads. Recent field studies have shown that an increased phytoplankton concentration stimulates both prespawning movement activity and spawning in males of the sea urchin S. intermedius (Zhadan, Vaschenko & Ryazanov, 2018). These findings indicate that a detailed quantitative analysis of the prespawning and spawning behaviors of sea urchins during mass spawning events is needed to understand whether the changes in movement activity may contribute to the enhancement of fertilization success. However, to date, there have been only rare and mostly descriptive observations of sea urchins’ mass spawning events under natural conditions.

The present study was intended to obtain quantitative data on the behavior of two cohabiting sea urchin species, Strongylocentrotus intermedius (A. Agassiz, 1864) and Mesocentrotus nudus (A. Agassiz, 1864) (= Strongylocentrotus nudus), during mass spawning events and to determine whether this behavior might contribute to the reproductive success of the studied species. To achieve this goal, we analyzed in situ video recording data of sea urchin behavior obtained during three spawning seasons (2017–2019). To increase the dataset, together with the data for the seasons 2017–2019, we additionally reanalyzed the data for the 2014–2016 seasons. We addressed the following: (1) how, and for how long before spawning, the behavior in two sea urchin species changes; (2) whether the distances between males and females change as a result of spawning behavior; (3) whether there is a link between spawned material and the changes in the prespawning and spawning behaviors of sea urchins; and (4) whether the behavior of nonspawning sea urchins changes during the spawning of conspecifics. We also compared the temporal characteristics of solitary and mass spawning and the dynamics of the numbers of spawners during mass spawning events in the two sea urchin species.

Materials and Methods

Study areas, sea urchins and video recordings

The field studies were conducted in two bays in the northwestern Sea of Japan, Kievka Bay (42.830°N, 133.691°E) and Alekseev Bay (42.981°N, 131.730°E). Six separate sets of observations were performed from 2014–2019, two in Kievka Bay (August–September of 2014 and 2015) and four in Alekseev Bay (May–June of 2016, July– September of 2017, June– September of 2018 and 2019). The objects of our studies were two sea urchin species, S. intermedius and M. nudus, which are common inhabitants of coastal communities in the study areas. The timing of the studies was chosen to coincide with the spawning seasons of these echinoids. For S. intermedius, it was May–September in Alekseev Bay and July–September in Kievka Bay (Zhadan, Vaschenko & Almyashova, 2017; Zhadan, Vaschenko & Ryazanov, 2018). The spawning season of M. nudus in both bays was July–August (P. Zhadan, 2020, personal communication; this study).

In Kievka Bay, the studies were performed at a depth of 6 m on relatively flat bedrock surrounded by large stones. In Alekseev Bay, the studies were performed at a depth of 2 m on a flat bottom covered with medium-sized gravel. To register the spawning behavior of sea urchins, continuous time-lapse video recording was performed with TLC200 Pro (Brinno Incorporated, Taipei City, Taiwan) video cameras, which were mounted on stanchions approximately 1 m above the bottom. The videos were taken in 1-min intervals at a resolution of 1,280 × 720 pixels. During the night, the cameras’ fields of view (approximately 1.0 × 1.5 m) were illuminated by LED lamps (1 W) synchronized with the cameras by a flash LED indicator, with impulse duration of 1 s.

To attract sea urchins to the cameras’ fields of view, flat mesh containers filled with the kelp Saccharina japonica (Laminariales, Phaeophyta), which is known to stimulate foraging activity in M. nudus and S. intermedius (Zhadan & Vaschenko, 2019), were used. Each container (hereafter the feeder) was composed of 2 steel frames with mesh stretched across them, 1.1 × 0.75 m in size, and contained 30–40 kg of kelp. In both study areas, the density of each sea urchin species was 1 to 2 ind. m−2 (Zhadan, Vaschenko & Ryazanov, 2018). In addition, 200–300 specimens of S. intermedius in Kievka Bay (2014, 2015) and 200–300 specimens of each species in Alekseev Bay (2016–2019) were collected within a 100- to 200-m radius of the cameras and carefully transplanted within an approximately 10-m radius. Sea urchins of both species themselves found the feeders, and within approximately 2 days, 30–70 individuals of each species populated the feeders. The feeders were replaced every 2–3 weeks when approximately 80% of laminaria was consumed by sea urchins. Sea urchins were carefully transferred from the old feeders to the bottom, and then they found themselves the feeders with fresh laminaria and populated them within several hours. It took approximately 5 min to change one feeder.

In both bays, four video cameras were used. In Kievka Bay, each camera was directed at the feeder. The entire area of the feeder and an additional 5–10 cm around its edges were in the cameras’ fields of view. In Alekseev Bay, where two feeders were located close to the vertical wall of the concrete pier (2016) or to a pyramid built of stones (2017–2019), two cameras were directed at the feeders, and two others were directed at the pier wall or the surface of the stone pyramid.

Video recording and statistical analyses

Just after the changes the feeders, isolated spawning males of M. nudus but not S. intermedius were noted on them. Therefore, to exclude the possibility that our manipulations could affect the results, video recordings taken for 8 h after the feeders were changed were not analyzed. During spawning, the sexes of both sea urchin species are easily distinguishable due to different colors of gametes: white in males and orange and light-yellow in females of S. intermedius and M. nudus, respectively. The recorded videos were viewed frame by frame, and the total number of specimens of each sea urchin species in the cameras’ fields of view and the numbers of spawning males and females were counted. In cases when at least 10 individuals took part in spawning for at least 1 h, we used the term “mass spawning”.

By means of the free software “Tracker” for video analysis (www.opensourcephysics.org/items/detail.cfm?ID=7365) we traced the spawning males and females with an interval of 1 min. The distance between two successive positions of the sea urchin (1 min apart) was defined as the step length (Lauzon-Guay, Scheibling & Barbeau, 2006; Zhadan & Vaschenko, 2019). The cell size (2 × 2 cm) of the feeder mesh was used as a scale. For males and females participating in mass spawning events, the tracing was started 60 min before the beginning of spawning of the first individual and finished when the last individual ceased to spawn (see Table S1 for raw data). Each track lasted approximately 180 min. For nonspawning sea urchins that were present on the feeders during mass spawning events, the duration of tracing was the same as that for spawners. In cases when sea urchins spawned alone or the spawners’ numbers were less than 10, the tracing began 60 min before the start of spawning of each spawner and lasted approximately 180 min.

Only full sea urchin tracks were included in the statistical analysis (see Table S1 for raw data). Exceptions included the tracks of sea urchins that spawned on the vertical surfaces of the pier wall or stone pyramid. In these cases, sea urchins before and during spawning moved strictly upward with a high speed and left the camera’s field of view before it was safe to conclude that they had finished spawning.

The beginning and end of the release of each gamete batch and duration of the intervals between the gamete batches were determined with an accuracy of 1 min. Total spawning duration was determined as a sum of total duration of gamete release and total duration of the intervals between the gamete batches (see Table S1 for raw data). In the cases when spawning of the specimen was recorded on only one frame, the individual spawning duration was considered to be 1 min. To analyze the sex-specific, species-specific and habitat-related differences in quantitative parameters of spawning (the release duration of one gamete batch, total duration of gamete release, time intervals between gamete batches, total duration of the intervals between the gamete batches and total duration of spawning) and to compare the time to reach the maximum numbers of spawning males and females participating in mass spawning events, the corresponding datasets were created and tested for adherence to a normal distribution (D’Agostino and Pearson omnibus normality test, P < 0.05). Since most of the datasets were not normally distributed, the nonparametric Mann–Whitney test, Wilcoxon matched-pairs signed-rank test and Spearman’s rank-order correlation were used. The same procedure was performed to compare temporal parameters of solitary and mass spawning in males of both sea urchin species.

To determine the moment of change (inflection point) in the locomotion rate of sea urchins before spawning, the 180-min time series of step lengths of spawning individuals were combined into 4 datasets, separately for each species and each sex, and aligned so that the beginnings of spawning of all individuals coincided. Within each dataset, the 60-min interval preceding the start of spawning was segmented by 11 possible inflection points (5 min apart). Each such point segmented the time-series dataset into two samples, one of which was assumed as the expected period before the beginning of change in sea urchin locomotion rate (“before start”) and the other as the expected period after the beginning of locomotion change (“after start”). Thus, 11 datasets were obtained, which were then compared using a nonparametric two-sample permutation test (Efron & Tibshirani, 1993). The test statistic Θ was estimated as the absolute difference between the medians, i.e., Θ = |Mebefore start – Meafter start|. The accumulated significance level (ASL, analogous to “P” of parametric tests) was determined as the proportion of cases when the absolute difference between two medians after data permutation (ΘPerm) was higher than our observed result (ΘExp). The first inflection point with ASL ≤ 0.01 was determined as a moment of statistically significant change in the locomotion rate of the sea urchins. The time interval preceding this point was defined as the “control interval”, and the following interval was defined as the “prespawning interval”, which ended with the onset of spawning. The 35-min interval after the beginning of spawning was defined as the “spawning interval”.

To analyze the sex-specific and species-specific differences in quantitative parameters of movement as well as to determine whether the locomotion rate in nonspawners changes during mass spawning events, the datasets on the step length of spawners and nonspawners of each sea urchin species measured during the control, prespawning and spawning intervals were pooled into corresponding datasets, checked for adherence to a normal distribution and tested with the Kruskal–Wallis test followed by Dunn’s multiple comparison test.

To determine whether sperm released by the males that began to spawn first during mass spawning events (hereafter “leading males”) influenced the locomotion rate of males that began to spawn 40 min later when there were already approximately 50% spawning males (hereafter “outsider males”), the step lengths of these two groups of males measured during the prespawning period were compared. The datasets on step lengths for leading males were supplemented with data on step lengths of males that spawned first in the cases that were not referred to as mass spawning events. After checking for adherence to a normal distribution, the data on step lengths of leading and outsider males of each sea urchin species were compared by the Mann–Whitney test.

To determine whether spawning males, females and nonspawning individuals come closer together during mass spawning, 2 analyses were performed. First, the distances from males, females and nonspawners to their common center of mass were determined. When calculating the center of the sea urchin mass, the conditional mass of males or females was taken as a multiple of the time of gamete release, and the conditional mass of nonspawners was taken as a value of 1. Second, the distances from each of the males and nonspawners to the center of mass of females were measured. Then, the differences between the distances measured 1 min apart during the 25-min control interval and during the spawning period, from the beginning of spawning to its completion in 95% of males, were determined, and the medians of these differences for the control and spawning periods were compared using the Mann–Whitney test. For graphical representation, the values of the displacement of each individual relative to its location at the beginning of the control interval were calculated 1 min apart. As these values were normally distributed, they are presented as the mean and 95% confidence interval.

Results

General characteristics of sea urchin spawning

In Kievka Bay in 2014–2015, spawning was recorded in 86 males and 6 females of S. intermedius and in 21 males of M. nudus. In Alekseev Bay in 2016–2019, spawning was recorded in 510 males and 19 females of S. intermedius and in 824 males and 39 females of M. nudus. Spawning occurred both in the presence and absence of spawning females. In the absence of females, the maximum numbers of S. intermedius and M. nudus spawning males were 11 and 20, respectively. For the entire duration of the observations, only a few females that spawned in the absence of males were recorded: 2 females of S. intermedius, which spawned on different days, and 2 females of M. nudus, which began to spawn with an interval of 4 min. The ratios of females to males for all spawning individuals were 1:24 for S. intermedius and 1:22 for M. nudus.

Calculations made from the total sea urchin number in the camera field of view, number of spawners and the female/male ratio show that males seem to be able to spawn several times during the spawning season. This is clearly demonstrated by a video recording obtained by one of the cameras, which in 2018 (June 11–17) recorded 3 mass spawning events with the participation of 41 males of M. nudus. Another 27 males spawned during the intervals between mass spawning events. Taking into account that the total sea urchin number in the camera field of view during this week decreased from 41 to 34 individuals and that the female/male ratio was 1:1, one may calculate that each of the males spawned approximately 3 times during this period. The total duration of sperm release during this week decreased by 3 times, but this change was not significant (Mann–Whitney test, U = 15, P = 0.08).

For each species, 12 mass spawning events were recorded, in which 10 and more individuals participated (Table 1). The percentage of spawning individuals ranged from 18% to 78% of the total number of sea urchins in the cameras’ fields of view. A total of 142 males and 15 females of S. intermedius and 183 males and 34 females of M. nudus participated in mass spawning. Thus, the numbers of S. intermedius and M. nudus males participating in mass spawning were 4 and 5 times lower and the numbers of females were 2.1 and 1.7 times lower, respectively, than the corresponding total numbers of spawning individuals. Both sexes participated in 8 of 12 spawning events for S. intermedius and in 10 of 12 spawning events for M. nudus. The ratios of females to males were 1:9 and 1:5 for S. intermedius and M. nudus, respectively.

Table 1 Locations and dates of mass spawning events in the sea urchins Strongylocentrotus intermedius and Mesocentrotus nudus.

Location	Species	Total number of sea urchins	Percentage of spawning sea urchins	Date	Number of full tracks of males/total number of males	Number of full tracks of females/total number of females	Sheet number in Table S1	
Kievka Bay	S. intermedius	24	43	16.08.2014	3/7	3/3	1	
Kievka Bay	S. intermedius	43	23	03.08.2015	9/10	0/0	2	
Kievka Bay	S. intermedius	29	39	03.08.2015	7/11	0/0	3	
Kievka Bay	S. intermedius	38	45	14.09.2015	12/16	1/1	4	
Kievka Bay	S. intermedius	56	18	16.09.2015	3/9	1/1	5	
Alekseev Bay	S. intermedius	45	22	17.05.2016	6/10	0/0	6	
Alekseev Bay	S. intermedius	65	26	19.05.2016	13/15	2/2	7	
Alekseev Bay	S. intermedius	72	28	20.05.2016	14/19	1/1	8	
Alekseev Bay	S. intermedius	66	33	22.05.2016	12/17	5/5	9	
AlekseevBay	S. intermedius	39	27	01.06.2016	5/9	1/1	10	
Alekseev Bay	S. intermedius	38	26	02.06.2016	6/10	0/0	11	
Alekseev Bay	S. intermedius	39	26	14.07.2018	5/9	1/2	12	
Alekseev Bay	M. nudus	56	52	13.07.2017	8/21	5/8	13	
Alekseev Bay	M. nudus	40	25	19.07.2017	6/10	0/0	14	
Alekseev Bay	M. nudus	38	53	21.07.2017	13/20	0/0	15	
Alekseev Bay	M. nudus	35	40	11.07.2018	7/11	2/3	16	
Alekseev Bay	M. nudus	41	27	11.07.2018	8/9	1/2	17	
Alekseev Bay	M. nudus	37	78	13.07.2018	18/23	5/6	18	
Alekseev Bay	M. nudus	63	41	13.07.2018	13/21	3/5	19	
AlekseevBay	M. nudus	57	18	17.07.2018	4/9	1/1	20	
Alekseev Bay	M. nudus	34	32	17.07.2018	8/9	2/2	21	
Alekseev Bay	M. nudus	37	49	16.08.2018	14/16	2/2	22	
Alekseev Bay	M. nudus	50	42	21.07.2019	10/18	2/3	23	
Alekseev Bay	M. nudus	47	38	26.07.2019	11/16	1/2	24	

Spawning occurred at dusk or at night in 19 of 24 mass spawning events recorded for both species (see Table S1). Twelve of 24 mass spawning events took place within the range from 0 to 4 days near the new moon, 2 events were recorded the day before the full moon, 3 events occurred when the moon was in waxing phase (first quarter), and 7 events were observed when the moon was in waning phase (third quarter) (see Table S1).

Despite the spawning seasons of S. intermedius and M. nudus overlap in the areas studied and the simultaneous spawning of several individuals of both sea urchins was sometimes observed, no one case of synchronous mass spawning with participation of 10 and more individuals of each species was recorded. However, on July 13, 2018, a case of simultaneous spawning of 29 individuals (23 males and 6 females) of M. nudus and 5 individuals (4 males and 1 female, i.e., < 10 individuals) of S. intermedius was recorded in Alekseev Bay (see Sheet 18 in Table S1). Оne can see that spawning of both sea urchin species is synchronous.

Temporal parameters of spawning

Males and females of both sea urchin species released gametes in batches. The number of batches in males varied from 1 to 10 in S. intermedius (median = 3) and from 1 to 9 in M. nudus (median = 2) and was significantly higher in S. intermedius (Mann–Whitney test, U = 4534, P = 0.0183). The numbers of batches in females of the two species did not differ significantly (range from 1 to 6, median = 2 for S. intermedius and range from 1 to 6, median = 1 for M. nudus, Mann–Whitney test, U = 138.5, P = 0.205). The release duration of one gamete batch (the time in which a gamete clot was present in the area near the gonopores), time intervals between gamete batches and total duration of spawning ranged widely (Table 2). Analysis of intersex differences showed that in both sea urchin species, the spawning duration of males was longer than that of females (P = 0.039 and P = 0.009 for S. intermedius and M. nudus, respectively) because although the total duration of gamete release did not differ significantly between sexes (P = 0.66 and P = 0.58), the total duration of intervals between gamete batches was significantly higher in males (P = 0.001 and P < 0.0001) (Table 2). Analysis of interspecific differences showed that in males, the release duration of one gamete batch, total duration of gamete release, total duration of intervals between gamete batches and total duration of spawning were significantly higher in S. intermedius than in M. nudus (Table 2). In females of the two sea urchin species, there were no significant differences in any of the spawning parameters studied.

Table 2 Temporal parameters of spawning of the sea urchins Strongylocentrotus intermedius and Mesocentrotus nudus participating in mass spawning events.

The data for 15 females (f) and 96 males (m) of S. intermedius and 24 females and 119 males of M. nudus are presented as the median and range (in parentheses). “n” is the number of measurements for each parameter. The Mann–Whitney test was used to reveal intersex and interspecies differences.

Parameter	S. intermedius	M. nudus	Intersex comparison	Interspecies comparison	
Male	Female	Male	Female	S. intermedius	M. nudus	Males	Females	
Release duration of one gamete batch (min)	3
(1–90)
n = 390	4
(1–34)
n = 38	2
(1–34)
n = 379	6
(1–21)
n = 50	m ≈ f
U = 6238;
p = 0.1040	m < f
U = 6148;
p = 0.0001	mint > mnud
U = 66350;
p = 0.0049	fint ≈ fnud
U = 869;
p = 0.4966	
Total duration of gamete release (min)	13.5
(1–92)
n = 96	14
(2–61)
n = 15	9
(1–62)
n = 119	14
(3–37)
n =24	m ≈ f
U = 683.5;
p = 0.6567	m ≈ f
U = 1611;
p = 0.5833	mint > mnud
U = 4847;
p < 0.0067	fint ≈ fnud
U = 181;
p = 0.5804	
Time interval between gamete batches (min)	4
(0–162)
n = 292	1.5
(0–35)
n = 28	4
(0–79)
n = 282	1
(0–11)
n = 37	m > f
U = 2862;
p = 0.0072	m > f
U = 2657;
p < 0.0001	mint ≈ mnud
U = 40895;
p = 0.8645	fint ≥ fnud
U = 401.5;
p = 0.1148	
Total duration of intervals between gamete batches (min)	15
(0–162)
n = 96	1
(0–50)
n = 15	9
(0–113)
n = 119	0
(0–11)
n = 24	m > f
U = 348.5;
p = 0.001	m > f
U = 871.5;
p < 0.0001	mint > mnud
U = 4760;
p = 0.0037	fint ≥ fnud
U = 162.5;
p = 0.062	
Spawning duration (min)	36
(1–188)
n = 96	24
(2–111)
n = 15	24
(1–140)
n =119	15
(1–48)
n = 24	m > f
U = 482;
p = 0.0398	m > f
U = 1178;
p = 0.0089	mint > mnud
U = 4423;
p = 0.0003	fint ≥ fnud
U = 162.5;
p = 0.1826	

Comparison of spawning parameters in S. intermedius sea urchins living in different bays showed that intersex differences were slightly more pronounced in sea urchins from Alekseev Bay (Table 3). In S. intermedius males from Alekseev Bay, all temporal parameters of spawning were significantly higher than those in Kievka Bay (Table 3). In S. intermedius females from different bays, there were no significant differences in any of the spawning parameters.

Table 3 Temporal parameters of spawning of the sea urchin Strongylocentrotus intermedius inhabiting different bays.

The data are presented as the median and range (in parentheses). “n” is the number of measurements for each parameter. The Mann–Whitney test was used to reveal the differences.

Parameter	Alekseev Bay	Kievka Bay	Intersex comparison	Interhabitat comparison	
Male	Female	Male	Female	Alekseev Bay	Kievka Bay	Males	Females	
Release duration of one gamete batch (min)	4
(1–90)
n = 257	5.5
(1–34)
n = 22	2
(1–13)
n = 133	3
(1–24)
n = 9	m < f
U = 2106;
p = 0.0449	m ≤ f
U = 2106;
p = 0.2847	mAlek > mKiev
U = 11580;
p < 0.0001	fAlek ≥ fKiev
U = 54.5;
p = 0.0517	
Total duration of gamete release (min)	25
(1–92)
n = 61	17
(5–61)
n = 10	6.5
(1–45)
n = 35	7
(2–24)
n =5	m ≈ f
U = 278;
p = 0.6101	m ≈ f
U = 85;
p> 0.9999	mAlek > mKiev
U = 433;
p < 0.0001	fAlek ≥ fKiev
U = 15;
p = 0.0856	
Time interval between gamete batches (min)	4
(0–162)
n = 201	2
(0–35)
n = 21	2
(0–35)
n = 91	1
(0–6)
n = 7	m > f
U = 1463;
p = 0.0192	m ≥ f
U = 157.5;
p = 0.0905	mAlek > mKiev
U = 6575;
p < 0.0001	fAlek ≥ fKiev
U = 401.5;
p = 0.2148	
Total duration of intervals between gamete batches (min)	23
(0–162)
n = 61	4.5
(0–50)
n = 10	7.5
(0–47)
n = 35	0
(0–11)
n = 5	m > f
U = 141.5;
p = 0.0048	m > f
U = 36.5;
p = 0.0337	mAlek > mKiev
U = 598.5;
p = 0.0004	fAlek ≥ fKiev
U = 50;
p = 0.2547	
Spawning duration (min)	59
(8–188)
n = 61	29
(5–111)
n = 10	20
(1–59)
n = 35	8
(2–24)
n = 5	m > f
U = 186;
p = 0.04354	m ≥ f
U = 55;
p = 0.2183	mAlek > mKiev
U = 395;
p < 0.0001	fAlek ≥ fKiev
U = 10.5;
p = 0.0799	

A comparison of the temporal parameters of solitary and mass spawning in males of S. intermedius and M. nudus from Alekseev Bay showed that in both species, the duration of release of one gamete batch and the total duration of gamete release were significantly higher during mass spawning events (Table 4). In S. intermedius males, the total duration of intervals between gamete batches and total spawning duration were also higher during mass spawning.

Table 4 Temporal parameters of solitary and mass spawning in males of the sea urchins Strongylocentrotus intermedius and Mesocentrotus nudus.

The data are presented as the median and range (in parentheses). “n” is the number of measurements for each parameter. The Mann–Whitney test was used to reveal differences between solitary and mass spawning.

Parameter	Strongylocentrotus intermedius	Mesocentrotus nudus	
Solitary spawning	Mass spawning	Statistics	Solitary spawning	Mass spawning	Statistics	
Release duration of one gamete batch (min)	2
(1–16)
n = 72	4
(1–90)
n = 257	U = 7024;
p = 0.0015	2
(1–12)
n = 52	3
(1–34)
n = 379	U = 8160;
p = 0.0396	
Total duration of gamete release (min)	13
(1–26)
n = 20	25
(4–92)
n = 61	U = 372.5;
p = 0.0069	6
(1–23)
n =20	9
(1–62)
n =119	U = 741;
p = 0.0021	
Time interval between gamete batches (min)	4
(0–48)
n = 36	4
(0–162)
n = 201	U = 3188;
p = 0.2551	4
(0–48)
n = 36	4
(0–79)
n = 282	U = 4916;
p = 0.7574	
Total duration of intervals between gamete batches (min)	6.5
(0–62)
n = 20	23
(0–162)
n = 61	U = 277;
p = 0.0001	7
(0–76)
n = 20	9
(0–113)
n = 119	U = 1186;
p = 0.5993	
Spawning duration (min)	20.5
(1–71)
n = 20	59
(8–188)
n = 61	U = 218;
p< 0.0001	15
(1–99)
n = 20	24
(1–140)
n = 119	U = 943.5;
p = 0.0588	

Dynamics of mass spawning

The spawning dynamics of males and females of both sea urchin species were generally similar: the number of spawners gradually increased, reaching a maximum, and then decreased to zero (Figs. 1A and 1B). The spawning males demonstrated similar dynamics both in the presence and absence of females (Figs. 1C and 1D). Figure 2 shows the dynamics of the total numbers of males and females of S. intermedius (Fig. 2A) and M. nudus (Fig. 2B) for all events of mass spawning. In most cases, the spawning duration of both sea urchin species was approximately 100 min (Figs. 2A and 2B). However, in 2 of 12 mass spawning events of S. intermedius, a second spawning wave took place, in which approximately 55% and 80% of females and males of the first spawning wave participated, respectively (Fig. 2A).

Figure 1 Examples of temporal dynamics of the numbers of simultaneously spawning sea urchins during mass spawning events.

(A and C) Strongylocentrotus intermedius. (B and D) Mesocentrotus nudus. (A and B) Spawning of sea urchins of both sexes. (C and D) Spawning of males in the absence of females. Blue and red lines indicate the numbers of males and females, respectively.

Figure 2 Temporal dynamics of the numbers of simultaneously spawning males and females of the sea urchins during mass spawning.

(A) Strongylocentrotus intermedius. (B) Mesocentrotus nudus. All data on the numbers of males and females participating in mass spawning events were combined into corresponding pools and aligned on the X-axis at the time point coinciding with the start of the first spawning in each mass spawning event (denoted by a vertical dotted line). Blue and red lines indicate the numbers of males and females, respectively.

In most mass spawning events of S. intermedius, the males started to spawn first (Fig. 2A). In 2 of 12 mass spawning events, S. intermedius females began to spawn first, but one of these cases was doubtful because when the female shifted 4 min after the start of spawning, the spawning male was found underneath. M. nudus males began to spawn first in all 12 mass spawning events (Fig. 2B). In all cases, males of both species finished to spawn after females (with exception of one case when M. nudus female finished to spawn simultaneously with males, see Sheet 24 in Table S1).

On average, S. intermedius and M. nudus females began to spawn 26.5 (–4–43) min and 24 (1–67) min (the data are presented as median and range) after the start of the first male spawning, respectively, when a great number of males had already spawned, and finished to spawn 12.0 (1–214) and 15.5 (0–70) min before the last male spawning (Figs. 1A, 1B, 2A and 2B). Although the females began to spawn later than the males, the times to reach the maximum numbers of spawning males and females did not differ significantly (Wilcoxon matched-pairs signed-rank test: n = 8, W = –18, P = 0.16 for S. intermedius and n = 10, W = –2, P = 0.95 for M. nudus). Spearman’s correlation analysis revealed significant positive relationships between the numbers of simultaneously spawning males and females for both species (Spearman’s r = 0.722 and 0.845 for S. intermedius and M. nudus, respectively, P < 0.0001).

Movement activity of sea urchins

Our data show that sea urchins of both species increased their movement activity shortly before and during spawning. A significant increase in step length in both sexes of both sea urchin species occurred from 30 to 35 min before the start of spawning (identified by the intersection of the threshold value of ASL = 0.01) (Figs. 3A–3D).

Figure 3 Temporal dynamics of the locomotion activity of males and females of the sea urchins before and during mass spawning.

(A and B) Strongylocentrotus intermedius. (C and D) Mesocentrotus nudus. The data on the length of sea urchins’ steps and the numbers of spawners were combined into corresponding pools and aligned on the X-axis at the time point coinciding with the beginning of spawning of each individual. Vertical dashed lines denote the boundaries of the control, prespawning and spawning intervals. The black solid line denotes the median step length of sea urchins. Vertical lines indicate the interquartile range (IQR). Blue and red lines indicate the numbers of spawning males and females, respectively (n).

During the prespawning interval, the average step length in males and females of both species was significantly higher than that in the control (Table 5). During the spawning interval, both sexes of M. nudus and males of S. intermedius accelerated even more, while S. intermedius females slowed down (Table 5). As the end of mass spawning approached, the movement rate in both sexes of both species gradually decreased (Figs. 3A–3D).

Table 5 Changes in the step length (cm) of the sea urchins Strongylocentrotus intermedius and Mesocentrotus nudus during different periods of mass spawning events.

The medians of step length were compared with the Kruskal–Wallis test followed by Dunn’s multiple comparison test. “n” is the number of full tracks of sea urchin movement; “ns”—not significant.

Interval	Statistical parameter	S. intermedius	M. nudus	Leading males	Nonspawning individuals	
Male	Female	Male	Female	S. intermedius	M. nudus	S. intermedius	M. nudus	
	n	96	15	119	24	21	33	85	50	
Control	Median	0	0.14	0.035	0	0	0	0.1	0.071	
Range	0–9.9	0–7.8	0–13.1	0–11.5	0–4.5	0–11.4	0–7.1	0–4.6	
Prespawning	Median	0.14	0.36	0.1	0.14	0.1	0.1	0	0	
Range	0–8.0	0–7.6	0–13.2	0–13.6	0–6.4	0–8.6	0–2.1	0–13.0	
Spawning	Median	0.32	0.27	0.47	0.42	0.45	0.42	0.1	0.1	
Range	0–14.0	0–7.0	0–12.2	0–14.2	0–8.0	0–13.3	0–10.78	0–9.5	
Prespawning/Control comparison	Mean rank diff.	519.1	262.7	443.6	207.6	133.4	237.1	–84.28	–4.1	
p	<0.0001	<0.0001	<0.0001	<0.0001	<0.0001	<0.0001	ns	ns	
Spawning/Control comparison	Mean rank diff.	1,869	131.8	243	505.7	520.3	670.7	336.3	161.9	
p	<0.0001	<0.0001	<0.0001	<0.0001	<0.0001	<0.0001	<0.0001	<0.001	
Spawning/Prespawning comparison	Mean rank diff.	1,350	−130.9	1,995	298	386.9	433.6	420.6	166	
p	<0.0001	<0.0001	<0.0001	<0.0001	<0.0001	<0.0001	<0.0001	<0.0001	

Leading males that began to spawn first during mass spawning events showed a significant increase in the average step length in both species during the prespawning period compared with the control, followed by further acceleration during the spawning interval (Fig. 4; Table 5). This finding indicates that accelerated locomotion of sea urchins before spawning resulted from external cues.

Figure 4 Temporal dynamics of the locomotion activity of males that started to spawn first during mass spawning (leading males).

(A) Strongylocentrotus intermedius. (B) Mesocentrotus nudus. The data on the step lengths of leading males and the numbers of spawners were combined into corresponding pools and aligned on the X-axis at the time point coinciding with the beginning of spawning of each male. Vertical dashed lines denote the boundaries of the control, prespawning and spawning intervals. The black solid line denotes the median step length of sea urchins. Vertical lines indicate the interquartile range (IQR). The blue line indicates the number of spawning males (n).

Comparison of the time series of step length of two groups of spawning males, leading males and outsider males, during the prespawning period showed no significant differences (Table 6). This result indicates that sperm of leading males did not influence the locomotion rate of outsider males during the prespawning period.

Table 6 Comparison of the step length (cm) of leading and outsider sea urchin males during the prespawning period.

The data are presented as the median and range (in parentheses) for 35-min intervals before the start of spawning. The Mann–Whitney test was used for comparison. “n” is the number of full tracks of sea urchin movement.

Species	Leading males	Outsider males	Statistics	
Strongylocentrotus intermedius	0.1
(0–6.4)
n = 21	0.14
(0–6.3)
n = 29	U = 355145, p = 0.0756	
Mesocentrotus nudus	0.1
(0–8.6)
n = 33	0.14
(0–8.4)
n = 30	U = 583245, p = 0.2407	

The number of females that spawned in the absence of other spawners was too small for statistical analysis (two S. intermedius females and two M. nudus females). However, it should be noted that in the only S. intermedius female, which undoubtedly began to spawn first during mass spawning, the average step length during the prespawning period was 3.9 times longer than that during the control period, and the average prespawning step length in two M. nudus females, which spawned in the absence of spawning males, was 2.1 times longer.

Analysis of the temporal dynamics of step length in nonspawning individuals showed that in both species, there was no significant difference in locomotion rate between the control and prespawning intervals, while a small but significant increase in this parameter occurred during the spawning interval (Fig. 5; Table 5).

Figure 5 Temporal dynamics of the locomotion activity of males (blue solid line) and nonspawners (green solid line) during mass spawning events.

(A) Strongylocentrotus intermedius. (B) Mesocentrotus nudus. The data on the step lengths of males and nonspawners were combined into corresponding time series and aligned on the X-axis at the time point coinciding with the beginning of spawning of the first individual in each mass spawning event. Vertical dashed lines denote the boundaries of the control, prespawning and spawning intervals. Vertical solid lines indicate the interquartile range (IQR).

Despite the absence of complete synchronization of the changes in locomotion rate, a significant positive correlation between the time series of step length of males and nonspawning individuals was found in 8 out of 12 mass spawning events for S. intermedius and in all 12 mass spawning events for M. nudus (ranges of Spearman’s r from 0.55 to 0.11 and P from <0.0001 to 0.31 for S. intermedius, Spearman’s r from 0.58 to 0.27 and P from <0.0001 to 0.004 for M. nudus).

Spatial distribution of sea urchin males and females during spawning

Measurement of the distances from males, females and nonspawners to their common center of mass revealed significant approach of sea urchins of both species during the spawning period (Fig. 6; Table 7). Measurement of the distances from males and nonspawners to the center of mass of females revealed significant approach during spawning in both sea urchin species (Fig. 7; Table 7). Figure 7 demonstrates the dynamics of the mean difference between the initial and measured 1-min interval distances from males and nonspawners to females. Since only one female took part in 5 of 8 mass spawning events of S. intermedius with the participation of both sexes, and in one case, two females were close to each other, the data in Fig. 7A mainly reflect a decrease in the distance from males and nonspawners to females. Since a larger number of females participated in the mass spawning events of M. nudus (Table 1), the data in Fig. 7B reflect a decrease in the distance from males and nonspawners to the area where females moved during spawning. For both sea urchin species, the displacement of males towards females was approximately two times higher than that of nonspawning individuals.

Figure 6 The changes in distances from spawners and nonspawners to their common center of mass during mass spawning of sea urchins.

When calculating the common center of mass, the conditional mass of spawning individuals was taken as a multiple of the time of gamete release. For nonspawners, the conditional mass was taken as a value of 1. The data for Strongylocentrotus intermedius and Mesocentrotus nudus are denoted by brown and black colors, respectively. X-axis: time of the sea urchin movement track (min). Left Y-axis: changes in the distances presented as the mean of differences between the initial and measured 1-min interval distances (cm) and 95% confidence intervals. Right Y-axis: data on the percentage of spawning males. Vertical dashed lines, from left to right, denote the boundary of the control interval, the time point when the first male in each mass spawning event began to spawn, and the time point when 95% of males spawned.

Figure 7 The changes in distances from males and nonspawners to the center of mass of females during mass spawning of the sea urchins.

(A) Strongylocentrotus intermedius. (B) Mesocentrotus nudus. When calculating the center of mass of females, female conditional mass was taken as a multiple of the time of gamete release. The data for nonspawners are denoted by a green color. X-axis: time of the sea urchin movement track (min). Left Y-axes: changes in distances presented as the mean of differences between the initial and measured 1-min interval distances (cm) and 95% confidence intervals. Right Y-axes: data on the percentage of spawning males. Vertical dashed lines, from left to right, denote the boundary of the control interval, the time point when the first male in each mass spawning event began to spawn, and the time point when 95% of males spawned.

Table 7 The distances (cm) from spawners and nonspawners to their common center of mass (CCM) and the center of mass of females (FCM) during sea urchin mass spawning.

The data are presented as the median and range (in parentheses). “n” is the number of full tracks of sea urchin movement.

Interval	Statistical parameter	Strongylocentrotus intermedius	Mesocentrotus nudus	
From all individuals to CCM	From males to FCM	From nonspawners to FCM	From all individuals to CCM	From males to FCM	From nonspawners to FCM	
	n	131	67	49	165	101	34	
Control	Median	17.16	20.50	23.74	19.15	25.06	30.18	
Range	1.15–36.91	4.96–46.89	5.92–50.50	0.74–58.80	3.07–68.16	2.58–72.04	
Spawning	Median	13.53	15.75	21.62	16.02	19.35	28.43	
Range	0.19–39.16	2.93–55.13	4.80–54.88	0.15–46.71	1.82–46.69	1.29–101.0	
Control/Spawning comparison	Mann–Whitney U	12,400,000	1,920,000	2,222,000	21,010,000	4,757,000	775,386	
p	<0.0001	<0.0001	<0.0001	<0.0001	<0.0001	=0.0007	
Number of values	12,746	4,962	5,220	11,106	7,699	3,058	

It should be noted that there was variability between mass spawning events in the median distances of males and nonspawners to the center of mass of females: the distances could decrease (in most cases), increase or remain almost unchanged (see Table S1 for raw data).

The spatial distribution of sea urchins during spawning on the vertical surface (pier wall) or inclined surface of stone pyramids was significantly different from that on flat food substrates. Males and females of both species moved strictly upward both before and during spawning. Spawning males, reaching the top of a large stone, continued to actively move along its surface in the absence of females. In cases when there was a spawning female nearby, the males slowed down or stopped. In contrast, females usually stopped active movement during spawning after reaching the top of the stone. S. intermedius females also displayed similar behaviors on food substrates when they were on top of other individuals. In M. nudus, the formation of such groups was not recorded in any of the mass spawning events.

Discussion

Despite the high density of sea urchins on food substrates, only 24 mass spawning events (each of 10 and more spawners) were recorded for both species over 6 spawning seasons (2014–2019), in which approximately 4 times fewer sea urchins were involved compared with the total number of spawners. This finding indicates that in S. intermedius and M. nudus, mass spawning is a relatively rare phenomenon that apparently occurs due to the complex interaction of both intrinsic (i.e., gonad maturity) and extrinsic (some environmental cue(s), primarily phytoplankton), factors. The application of continuous around-the-clock time-lapse video recordings and feeders with kelp attractive to sea urchins allowed us to perform a high-temporal-resolution (at 1-min intervals) quantitative analysis of the behaviors of S. intermedius and M. nudus before and during mass spawning events under natural conditions. To the best of our knowledge, this is the first such analysis to examine the representatives of mobile broadcast invertebrates with planktotrophic larvae. The high temporal resolution of the video recording method provides an opportunity for the mutual interpretation of results from laboratory and field studies, and long-term around-the-clock observations allowed us to replenish the data on the behavior of broadcast invertebrates during mass spawning obtained in situ by diving.

Temporal characteristics and quantitative dynamics of mass spawning are similar in S. intermedius and M. nudus

Males and females of both sea urchin species exhibited an intermittent (“pulse”) pattern of spawning, as shown earlier for S. intermedius (Zhadan, Vaschenko & Ryazanov, 2018). The most significant difference in temporal parameters of spawning was the longer duration of male spawning in terms of both intersex comparison (in both species, males spawned longer than females) and interspecies comparison (S. intermedius males spawned longer than M. nudus males). This result is consistent with the conclusion of Lotterhos & Levitan (2010) based on an analysis of data reported in studies on spawning duration in males and females of 13 taxonomic groups of broadcast invertebrates, including echinoderms. However, in this review, the spawning process of three sea urchin species (M. franciscanus, Strongylocentrotus droebachiensis and Diadema antillarum) was characterized as continuous (“plume”) based on occasional diving observations. In our studies, along with the intermittent (“pulse”) pattern of spawning, we showed that in S. intermedius and M. nudus, the longer spawning duration of males was due to the longer pause between sperm batches, while there were no sex differences in the total duration of gamete release.

Taking into consideration that the duration of gamete release in our study was assessed by the presence of gamete clot on the aboral surface of sea urchin test, a question may arise whether different spawning duration of males and females may be due to different physical properties of sperm and eggs and the influence of water advection on gametes’ dispersion. To date, only a few studies addressed these issues in sea urchins (Thomas, 1994; Yund & Meidel, 2003; see also Crimaldi & Zimmer, 2014 for review). Based on the comparison of the retention times of eggs and sperm on the tests of three morphologically different sea urchin species (Tripneustes gratilla, Echinometra mathaei and Colobocentrotus atratus) under different water velocity (Thomas, 1994) and our results on temporal parameters of S. intermedius and M. nudus spawning, we may suggest that the earlier cessation of spawning in females of S. intermedius and M. nudus compared to males during mass spawning events is due to longer spawning duration of males but not different physical properties of female and male gametes.

In both S. intermedius and M. nudus, the temporal dynamics of the numbers of males and females participating in mass spawning were well synchronized, so that the maximum numbers of simultaneously spawning sexes coincided. Currently, to the best of our knowledge, there is surprisingly little information on the temporal patterns of spawning in situ both for sea urchins and other groups of broadcast spawners. In terms of quantification of the dynamics of mass spawning, the studies of in situ spawning of the sea cucumber Cucumaria frondosa, the echinoderm species with long-lived planktonic lecitothrophic larva, seem to be the most comprehensive (Hamel & Mercier, 1995, 1996). Mass spawning event started from the spawning of isolated males. The number of males reached maximum (83%) after 10 h, whereas maximum number (87%) of females was recorded after 12 h when the males’ number was less than 32%. Thus, the maxima of spawning activity in males and females of C. frondosa did not coincide. However, this delay between the peaks of male and female spawnings could have a benefit for fertilization success because it allows to attain a maximum concentration of sperm in the water column prior to female spawning (Hamel & Mercier, 1996). We believe that the differences in the temporal dynamics of the numbers of spawners of C. frondosa (Hamel & Mercier, 1995) and S. intermedius and M. nudus (our study) during mass spawning can be explained by interspecific variations in reproductive physiological adaptations and behavior associated with different life-history models and aimed at optimization of gamete dispersion and fertilization success in each species.

Generally, males of S. intermedius and M. nudus exhibited much more active spawning behavior than females. First, they began to spawn earlier and ended spawning later than females in all cases except for one mass spawning event when one S. intermedius female undoubtedly began to spawn before males. Moreover, males of both sea urchin species were capable of spawning in the absence of females, demonstrating temporal dynamics of spawner numbers similar to those during mass spawning with the participation of both sexes. In contrast, females began to spawn after several males had already spawned, and over six spawning seasons, for each species, only two cases were recorded as not belonging to mass spawning when females spawned in the absence of males. Our observations support the currently accepted view that in broadcast spawners, males predominantly spawn before females (see Levitan, 1998; Mercier & Hamel, 2009; Thorson, 1950 for review). For sea urchins, rare exceptions have been encountered in field studies, for example, spawning of S. droebachiensis females in the absence of males (Pearse et al., 1988).

Second, the numbers of males participating in mass spawning were significantly higher than those of females (the female/male ratios were 1:9 and 1:5 for S. intermedius and M. nudus, respectively). This is significantly different from the sex ratio in the natural populations of these species, which is close to 1:1 (Zhadan, Vaschenko & Ryazanov, 2018; P. Zhadan, 2020, personal communication). Similar female/male ratios during mass spawning in the field were found for other sea urchin species, such as Strongylocentrotus purpuratus (1:4, Levitan, 2002), M. franciscanus (1:8, Levitan, 2002) and S. droebachiensis (1:4, Himmelman et al., 2008).

The significant excess of spawning males over the number of spawning females can be explained in several ways. First, males develop the ability to release gametes earlier than females and retain this ability after the completion of female spawning, as was shown for S. intermedius (Zhadan, Vaschenko & Almyashova, 2015; Zhadan, Vaschenko & Ryazanov, 2018). The same sexual maturity pattern was reported for males of the sea urchin Lytechinus variegatus (Reuter & Levitan, 2010). Second, males seem to be able to spawn several times during the spawning season (Zhadan, Vaschenko & Ryazanov, 2018; this study). Third, males seem to be more sensitive to external triggers of spawning than females. Consistent with this suggestion, laboratory experiments demonstrated the higher sensitivity of males to external stimuli such as phytoplankton and sperm for the sea urchins S. droebachiensis and L. variegatus, mussel Mytilus californianus and crown-of-thorns starfish Acanthaster cf. solaris (Caballes & Pratchett, 2017; Reuter & Levitan, 2010; Starr, Himmelman & Therriault, 1990).

As proposed previously by several researchers, the dynamics of the numbers of spawners during mass spawning of sea urchins are likely to form due to the presence of positive feedback (Reuter & Levitan, 2010; Starr, Himmelman & Therriault, 1990). Thus, spawning products of the individuals most sensitive to environmental factors stimulate the spawning of neighbors, as shown in laboratory experiments in which sperm alone induced spawning in L. variegatus (Reuter & Levitan, 2010) and a combination of phytoplankton and sperm had a synergistic effect on spawning induction in S. droebachiensis females (Starr, Himmelman & Therriault, 1992). It should be noted, however, that in the other laboratory and field experiments, no effect of water-borne gametes of L. variegatus on spawning induction in sexually mature conspecifics of both sexes was found (McCarthy & Young, 2004). Our results showed that in both S. intermedius and M. nudus, the duration of sperm release during mass spawning events was longer than that of solitary spawning. These findings indicate, on the one hand, that during mass spawning, fertilization success may be enhanced by not only the larger number of spawners but also the higher intensity of gamete release. On the other hand, our results suggest that (1) water-borne gametes cannot be considered a primary cue for the induction of mass spawning and (2) one of the reasons for the development of mass spawning events in broadcast spawners may be favorable environmental conditions for the stimulation of spawning, for example, phytoplankton concentrations higher than those during solitary spawning. Our previous field studies showed that an increase in phytoplankton concentration triggers spawning in natural S. intermedius populations (Zhadan, Vaschenko & Almyashova, 2015; Zhadan et al., 2016; Zhadan, Vaschenko & Ryazanov, 2018). The timing, duration and completeness of spawning at the population level clearly depended on phytoplankton abundance during the spawning season. In the present study, we showed that the temporal parameters of spawning in S. intermedius males were different in bays with different levels of phytoplankton. In the bay with the higher phytoplankton level, S. intermedius males exhibited significantly longer durations of gamete release, intervals between gamete batches and total spawning processes. Considering that most mass spawning events in S. intermedius and M. nudus occurred at night and close to new or full moon phases, the nighttime and lunar phases may be additional environmental factors increasing the probability of mass spawning.

In the present work, we did not include data on the role of phytoplankton in the stimulation of spawning in M. nudus. Our preliminary data provide evidence that an increased concentration of phytoplankton triggers spawning in the M. nudus population (P. Zhadan, 2020, personal communication). However, the relationships between spawning and environmental factors for this species are more complex than those for S. intermedius. At a low concentration of phytoplankton in the environment, mature gametes accumulated in M. nudus gonads and were released during storm events. These data will be the subject of a future article.

Sea urchins increase movement activity before and during mass spawning

In a previous study, we reported that S. intermedius males began to move actively just before spawning and retained this activity during the spawning process (Zhadan, Vaschenko & Ryazanov, 2018). In the present work, a precise time was determined when an increase in the locomotion rate of sea urchins happened before spawning. In both sexes of S. intermedius and M. nudus, this phenomenon occurred 35 min before the start of spawning. M. nudus males and females and S. intermedius males gradually increased the locomotion rate right up to the time of spawning, while S. intermedius females reached a plateau in the average step length approximately 15 min before the beginning of spawning and exhibited a lower locomotion rate during spawning than males (Fig. 3; Table 6). Many S. intermedius females stopped moving after climbing to the top of a stone or to another individual. These results are consistent with our previous conclusion about the increased movement activity of S. intermedius males during spawning (Zhadan, Vaschenko & Ryazanov, 2018).

Another important finding of this study was that sea urchins that started to spawn first during mass spawning (leading males) as well as males that spawned alone increased their movement rate before spawning in the absence of other spawning individuals. This observation clearly indicates that the increase in locomotion activity in S. intermedius and M. nudus before spawning is due to environmental factor(s). Moreover, we found no differences in the prespawning locomotion rate between leading males and outsider males, which started to spawn when the sperm of leading males was already present in the environment. Nevertheless, this finding cannot exclude the influence of sperm as a factor contributing to the stimulation of spawning in sea urchins. Our results showing that nonspawning individuals of both species increased locomotion activity in the presence of spawning males and females suggest a possible effect of released gametes on the movement activity of nonspawners during mass spawning events. This suggestion is congruent with the results of laboratory experiments in which phytoplankton and/or sperm treatments stimulated spawning behavior defined as climbing up the sides of the beaker in L. variegatus males and females, whereas only sperm and the combination of phytoplankton and sperm stimulated such a behavior in sexually immature individuals (Reuter & Levitan, 2010).

Simultaneously, we observed that in some cases, nonspawners began to actively move before their neighbors started to spawn. Sheet 34 in Table S1 demonstrates that on the nonfood substrate where there were 4 M. nudus individuals, 2 females and 2 nonspawners, the nonspawners began to actively move before the females began to spawn. Moreover, we also observed that in September, after all M. nudus individuals had already spawned, with an increase in the concentration of phytoplankton up to 5–10 μg l–1, sea urchins moved from the feeders to a stone pyramid, and with a decrease in phytoplankton concentration, they returned again to the food substrate. These observations suggest that immature sea urchins have the ability to perceive some environmental signal(s), in the case of their high intensity, which stimulates spawning behavior.

Sea urchin males and females become closer during mass spawning

One of the most interesting questions concerning the reproductive behavior of sea urchins, to which there is still no answer, is the question of whether sea urchins approach each other during spawning. On flat food substrates, the movement of spawning males of S. intermedius and M. nudus appeared multidirectional (Zhadan, Vaschenko & Ryazanov, 2018; this study). However, comparison of the distances between males and females measured during the control and spawning periods revealed that during spawning, males and females of both sea urchin species became significantly closer. Moreover, nonspawners also approached females, although this approach was less pronounced than that between males and females.

To date, due to rare direct field observations of sea urchin spawning, information on the spatial distribution of sea urchin species before and during spawning has been scarce. The sea urchin S. droebachiensis formed no spawning aggregations during echinoderm mass spawning events off the Mingan Islands in the northern Gulf of St. Lawrence in eastern Canada (Himmelman et al., 2008), whereas sea urchins T. gratilla off the island of Maui (Hawaii) spawned within small groups of 2–5 (Johnson & Ranelletti, 2017). One case of L. variegatus spawning in an aggregation of several hundred sea urchins was observed in St. Joseph Bay, Florida (Reuter & Levitan, 2010). To the best of our knowledge, the study reported by Levitan (2002) is the only work where nearest-neighbor distances between spawning and nonspawning M. franciscanus sea urchins during mass spawning events in the field were determined. Based on the data on the distribution of sea urchins that were mapped at 30-min intervals for 2 h, it was concluded that spawners and nonspawners did not become more aggregated during spawning. In our study, sea urchin trajectories were traced at 1-min intervals for 3 h, and the changes in distances from the mass centers of females to males and nonspawners were determined. This approach allowed us to show that sea urchins of S. intermedius and M. nudus did not form true aggregations before and during mass spawning events, but they undoubtedly approached each other. The most significant changes in the distances between spawning and nonspawning S. intermedius and M. nudus sea urchins occurred during the first 50–60 min after the beginning of mass spawning (Fig. 7), and the ranges of these changes varied widely, as males/nonspawners both approached the females and moved away (Table 7). Therefore, the time resolution of 30 min (Levitan, 2002) did not seem to be sufficient to detect the changes in nearest-neighbor distances between sea urchins during spawning.

Since fertilization success in sea urchins depends on the distance between the male and female (Levitan, 2002, 2005), their approach during mass spawning is important for reproductive success. The meaning of the approach between nonspawning individuals and females is not very obvious. In this regard, a question arises concerning whether spawning males and nonspawning individuals exhibit similar behaviors due to their responses to some chemical cues (pheromones) that are released together with gametes shed by conspecifics, most likely females. The last supposition is based on our observations that despite females moving more actively than males immediately before spawning, they usually stopped after climbing the elevated surface, followed by the start of spawning. Simultaneously, after the beginning of spawning, males accelerated in the absence of females but stopped near females.

Both experimental and field studies show that sea urchins use chemosensation to avoid predators as well as to find food (Campbell et al., 2001; Mann et al., 1984; Spyksma, Taylor & Shears, 2017; Zhadan & Vaschenko, 2019). Genomic analysis suggests that an elaborate chemosensory system involving several hundred putative chemoreceptor genes, in particular those encoding olfactory receptors, operates in sea urchins (Burke et al., 2006; Raible et al., 2006). However, to date, there is no information on the presence and chemical nature of pheromones in sea urchins. Recently, mass spectrometry, genomic and proteomic analyses were performed to identify the protein composition of water-borne plumes released from aggregating spawning crown-of-thorn starfish (Acanthaster planci) (Hall et al., 2017). The proteins secreted by A. planci have been shown to consist largely of signaling factors and hydrolytic enzymes that may be detected by putative olfactory receptors expressed in the external tissues of starfish. We believe that the new data on the patterns of sea urchin locomotion activity during mass spawning obtained in the present study will serve as the physiological basis for the search for biomolecules that may play a role in pheromones in this group of echinoderms.

Conclusion

Our results show that two sea urchin species with planktotrophic larvae, S. intermedius and M. nudus, which were monitored in their natural environment, display similar behaviors during mass spawning events. Males and females of both species responded to some environmental cue(s), most likely phytoplankton, by increasing their locomotion rate 35 min before the start of spawning. Subsequently, they accelerated until the beginning of and during spawning. Males appeared to be more sensitive to external trigger(s) of spawning than females; therefore, during mass spawning events they began to actively move earlier, in the absence of other spawners. Nonspawners of both species also increased their locomotion activity, but at a later time, in the presence of spawning males and females, and less prominently than spawners. On a vertical surface, both echinoids moved strictly upward, whereas on a flat food substrate, their movement was multidirectional. Spatial distribution analysis showed that although neither echinoid formed spawning aggregations on flat surfaces, the males and, to a much lesser extent, nonspawners approached females during mass spawning.

The temporal dynamics of the numbers of males and females participating in mass spawning were well synchronized in both echinoids so that the maximum numbers of simultaneously spawned sexes coincided. However, males of both sea urchin species exhibited much more active spawning behavior than females: (1) males began to spawn earlier and ended spawning later than females; (2) the spawning duration of males was longer due to the longer pause between sperm batches; and (3) males seemed to be able to spawn several times during the spawning season.

Temporal and quantitative patterns of behavior of the sea urchins S. intermedius and M. nudus before and during mass spawning may be considered a set of behavioral adaptations aimed at increasing fertilization success. The high sensitivity of males to environmental factor(s), primarily phytoplankton, appears to be a large-scale adaptation characteristic for many broadcast spawners with planktotrophic larvae and is apparently one of the prerequisites for the development of mass spawning events. The nighttime and new and full moon phases apparently to be modulating factors increasing the probability of mass spawning. The longer spawning duration in males compared with females, longer duration of sperm release during mass spawning events compared with that during solitary male spawning, longer durations of sperm release and total time of spawning in males inhabiting the bay with higher levels of phytoplankton and approach of males and females during mass spawning may be considered small-scale adaptations that promote the likelihood of fertilization.

Supplemental Information

Supplemental Information 1 Parameters of spawning events in the sea urchins Strongylocentrotus intermedius and Mesocentrotus nudus.

Click here for additional data file.

We thank M.Yu. Cheranev for the help in making the underwater installation for video recording and L.Yu. Pavin for assistance in the field.

Additional Information and Declarations

Competing Interests

Author Contributions

Data Availability

The authors declare that they have no competing interests.

Peter M. Zhadan conceived and designed the experiments, performed the experiments, analyzed the data, prepared figures and/or tables, authored or reviewed drafts of the paper, and approved the final draft.

Marina A. Vaschenko analyzed the data, prepared figures and/or tables, authored or reviewed drafts of the paper, and approved the final draft.

Peter A. Permyakov analyzed the data, prepared figures and/or tables, and approved the final draft.

The following information was supplied regarding data availability:

Raw measurements are available as a Supplemental File.

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
