# Peer review of "Quantitative study of the behavior of two broadcast spawners, the sea urchins Strongylocentrotus intermedius and Mesocentrotus nudus, during mass spawning events in situ"

_PeerJ, doi:10.7717/peerj.11058_

## Round 0.1 · original submission · Major Revisions

· Academic Editor

Major Revisions

Two expert reviewers have evaluated your manuscript and their comments can be seen below. Both offer very helpful advice on improvements to be made to the mansucript. Please follow their advice in a revision of the manuscript.

Reviewer 1 ·

Basic reporting

No major issues

Experimental design

No major issues

Validity of the findings

No major issues

Additional comments

This is an interesting and informative study of patterns of behavior and gamete release in two species of sea urchins. Although there are many papers that report patterns of natural spawning, few report long-term quantitative data. As such this will be a welcome addition to the literature on spawning in broadcast spawning invertebrates.

There are only a few things that the authors might consider in a revision.
1. The control for behavior appears to be the period prior to spawning. Why not take a parallel data set from time periods picked at random on days in which no spawning was observed?
2. The data on spawning duration appears to be video frames in which gametes are noted on the aboral surface. Gametes often pool on the surface of sea urchins and the duration they remain on the surface is likely influenced by water flow (advection and turbulence). Is there any information on whether gametes tend to remain on the aboral surface was influenced by flow and whether eggs and sperm respond to flow in a similar manner?
3. Figure 1 suggests that males generally do initiate spawning prior to females. At the end of the event the males and females appear to stop spawning at around the same time (males might be active for slightly longer). It might be useful to test to see the asynchrony (males more likely to spawn) is symmetrical (female spawning is centered at the mean male spawning time) or if the overlap in spawning time between sexes is shifted to the end of the spawning time (both sexes tend to stop at around the same time).
4. The authors are very thorough on incorporating the literature on spawning. I would suggest they consider an often overlooked paper by Hamel and Mercer (1996 Beche-de-mer inf. Bull. 8:34-40). This is a very detailed and quantitative investigation of the temporal pattern of spawning. There is also a pretty extensive literature on spawning observations of sessile invertebrates (e.g. corals). If the authors thought this was too much to cover, they might simply refer to this literature in a more general way so that readers are aware of the quantitative assessment of this group of (often) broadcast spawners.
5. The spawning seasons of these two species overlap, it might be good to add a few sentences on the degree to which they did (or did not) both species spawned synchronously. In other studies, Strongylocentrotus and Mesocentrotus were observed to spawn together.

·

Basic reporting

General Comments

The manuscript describes a series of observations based on video recordings during the spawning season of 2 species of sea urchins, Strongylocentrotus intermedius and Mesocentrotus nudus. The information provided is particularly useful to marine ecologist to get a better a understanding of spawning behavior of this species of sea urchins under natural conditions and also, could be useful to Aquaculturist to help. Them manipulate spawning of sea urchins in laboratory conditions. Considering the difficulties to get in situ observations of spawning of sea urchins, the results provided add to the existing information on the subject for those particular species. My main observations to the manuscript are the following:
1. The introduction is too long. I do not think is necessary to include a big number of references to support the text. Considering that the work is not a review, I consider would be enough to select few of them, maybe the ones that the authors think they are more relevant and more closely related to the species of sea urchins studied. Also, I do not think is necessary to include the spawning behavior of other classes of equinoderms different that sea urchins. There are several differences among them and from my point of view, these comparisons are not relevant for the study and only contribute to make a very long introductory section.
2. The Discussion is a mixture of results already included (or at least they should be included in the Results section) and material already mentioned in the Introduction (or should be included if necessary), combined with a more proper Discussion.
I consider the authors should try to avoid to repeat introductory or results material and focus their Discussion in highlight the main findings of the study and contrasting to relevant literature. As in the Introduction, they have the tendency to used results that were already mention, to support their discussion. The following is an example to illustrate my point:

14. Lines 609-611. The authors don’t need to repeat information that already was mentioned in other sections (in this case Methods). The following text “In our study, sea urchin trajectories were traced at 1-min intervals for 3 h, and the changes in distances from the mass centers of females to males and non spawners were determined.” were already mention in methodology or should be included.
The same is true for the following text, already mentioned in the Results section. “The most significant changes in the distances between spawning and nonspawning S. intermedius and M. nudus sea urchins occurred during the first 50–60 min after the beginning of mass spawning (Fig. 7), and the ranges of these changes varied widely, as males/nonspawners both approached the females and moved away (Table 7). (Lines 614-617 in the original manuscript).

Finally, considering the difficulties and the logistics to capture the right moments of spawning of the species studied under their natural conditions in the ocean, the information provided I consider is very valuable; however, the authors did some manipulations of the natural conditions (moving organisms and providing food to set up the initial conditions on the sites), that maybe had an effect on the behavior of the species. I consider they at least, speculate in the Discussion if these handling of the species could have an impact in their spawning behavior or justify why they think this initial manipulation is irrelevant.
Detail comments about the particular sections of the manuscript are in the following paragraphs.


Basic Reporting

1. Lines 78-85. I do not think is necessary to use that many number of references in lines 78-85, to support their statements about environmental factors that influence spawning behavior in sea urchins. Chose the ones that consider most representative of the existing literature and I don’t think is necessary to elaborate about spawning behavior in other classes of echinoderms including ofiuroids, asteroids or holoturoids, considering that we know they are different from sea urchins in several ways.

Experimental design

5. Lines 233-234. Is there a possibility that the spawning duration was longer?

6. Lines 267-271. Suggestion: This paragraph could be integrated with little changes to the proceeding one (Lines 262-266), considering that both ends with the same statistical analysis to both sets of data.

They also need to clarify in the Discussion if the initial manipulation of the organisms and the provision of food could alter their spawning behavior

Validity of the findings

7. Line 317-322. Could you please indicate here in the text with more detail the interval of time at which mass spawning occurred during dusk or night and how many days before or after the new or full moon?

8. Lines 321-322. Did you performed any correlation analysis that support this phrase “there was no evident link to any lunar phase” (Lines 321-322), considering that at the beginning of the paragraph is mentioned that “spawning events occurred close to new od full moon”

9. Lines 366-367. If you are talking about an “average”, it is not clear to me what measure of variation represents the “±” associated to the average. Please specify if it is a confident interval or Standard Deviation or Error, or any other measure of variability.

10. Lines 375-376. Be more direct to describe the findings. Instead of saying, “When watching video records, it was clearly seen that sea urchins of both species increased their movement” (We already know you were watching video records, that is the basis of the whole study). Directly say: “Our data show that sea urchins of both species increased their movement…”

11. Line 490. S franciscanus is now Mesocentrotus franciscanus

12. Lines 498-504. This whole paragraph belongs to the results section: “The last observation is clearly demonstrated by a video recording obtained by one of the cameras, which in 2018 (June 11–17) recorded 3 mass spawning events with the participation of 41 males of M. nudus. Another 27 males spawned during the intervals between mass spawning events. Taking into account that the total sea urchin number in the camera field of view during this week decreased from 41 to 34 individuals and that the female/male ratio was 1:1, one may calculate that each of the males spawned approximately 3 times during this period. The total duration of sperm release during this week decreased by 3 times, but this change was not significant (Mann–Whitney test, U = 15,P = 0.08).”
The authors don’t need to repeat what they already mention in the Results section. The only phrase they need here to support their second observation [(Text From the original manuscript, lines 497-498: “Second, males seem to be able to spawn several times during the spawning season (Zhadan, Vaschenko & Ryazanov, 2018; this study)]” is “The last observation is clearly demonstrated by our recording videos where several mass spawining of males were recorded (see results of the one in June 11-17, 2018).

13. Lines 537-540. The authors mention that “Considering that most mass spawning events in S. intermedius and M. nudus occurred at night and close to new or full moon phases, the night time and lunar phases may be additional environmental factors increasing the probability of mass spawning. However, in their Results section, they mention that “there was no evident link to any lunar phase” (Lines 321-322 of the original mansucript). Could you please clarify this or elaborate according to your results. Maybe if they analyze the number of nights (before New or Full Moon), when the spawning occurs, they could find a possible pattern related to the phases of the moon. This has been well documented for example in many coral species.

14. Lines 609-611. The authors don’t need to repeat information that already was mentioned in other sections (in this case Methods). The following text “In our study, sea urchin trajectories were traced at 1-min intervals for 3 h, and the changes in distances from the mass centers of females to males and non spawners were determined.” were already mention in methodology or should be included.
The same is true for the following text, already mentioned in the Results section. “The most significant changes in the distances between spawning and nonspawning S. intermedius and M. nudus sea urchins occurred during the first 50–60 min after the beginning of mass spawning (Fig. 7), and the ranges of these changes varied widely, as males/nonspawners both approached the females and moved away (Table 7). (Lines 614-617 in the original manuscript).

Additional comments

General Comments

The manuscript describes a series of observations based on video recordings during the spawning season of 2 species of sea urchins, Strongylocentrotus intermedius and Mesocentrotus nudus. The information provided is particularly useful to marine ecologist to get a better a understanding of spawning behavior of this species of sea urchins under natural conditions and also, could be useful to Aquaculturist to help. Them manipulate spawning of sea urchins in laboratory conditions. Considering the difficulties to get in situ observations of spawning of sea urchins, the results provided add to the existing information on the subject for those particular species. My main observations to the manuscript are the following:
1. The introduction is too long. I do not think is necessary to include a big number of references to support the text. Considering that the work is not a review, I consider would be enough to select few of them, maybe the ones that the authors think they are more relevant and more closely related to the species of sea urchins studied. Also, I do not think is necessary to include the spawning behavior of other classes of equinoderms different that sea urchins. There are several differences among them and from my point of view, these comparisons are not relevant for the study and only contribute to make a very long introductory section.
2. The Discussion is a mixture of results already included (or at least they should be included in the Results section) and material already mentioned in the Introduction (or should be included if necessary), combined with a more proper Discussion.
I consider the authors should try to avoid to repeat introductory or results material and focus their Discussion in highlight the main findings of the study and contrasting to relevant literature. As in the Introduction, they have the tendency to used results that were already mention, to support their discussion. The following is an example to illustrate my point:

14. Lines 609-611. The authors don’t need to repeat information that already was mentioned in other sections (in this case Methods). The following text “In our study, sea urchin trajectories were traced at 1-min intervals for 3 h, and the changes in distances from the mass centers of females to males and non spawners were determined.” were already mention in methodology or should be included.
The same is true for the following text, already mentioned in the Results section. “The most significant changes in the distances between spawning and nonspawning S. intermedius and M. nudus sea urchins occurred during the first 50–60 min after the beginning of mass spawning (Fig. 7), and the ranges of these changes varied widely, as males/nonspawners both approached the females and moved away (Table 7). (Lines 614-617 in the original manuscript).

Finally, considering the difficulties and the logistics to capture the right moments of spawning of the species studied under their natural conditions in the ocean, the information provided I consider is very valuable; however, the authors did some manipulations of the natural conditions (moving organisms and providing food to set up the initial conditions on the sites), that maybe had an effect on the behavior of the species. I consider they at least, speculate in the Discussion if these handling of the species could have an impact in their spawning behavior or justify why they think this initial manipulation is irrelevant.

---

## Round 0.2 · accepted · Accept

· Academic Editor

Accept

I am satisfied with the changes made to the manuscript.

·

Basic reporting

Agree with modifications provided by the authors in the revised version

Experimental design

Agree with modifications provided by the authors in the revised version

Validity of the findings

Agree with modifications provided by the authors in the revised version

Additional comments

I consider the additions and modifications provided in the revised version of the original manuscript clarify my comments to the original manuscript and consider the information provided by the authors a good contribution to get a better understanding of spawning behavior in sea urchins under natural conditions were always is a logistic challenge to get the information. I consider the manuscript ready to be published, after the final revision of format, if any, by the Journal.